# Understanding the effects of early degradation on isotopic tracers: implications for sediment source attribution using compound-specific isotope analysis (CSIA)

Pranav Hirave[1], Guido L.B. Wiesenberg[2], Axel Birkholz[1], Christine Alewell[1]

[1]Environmental Geosciences, Department of Environmental Sciences, University of Basel, Basel, Switzerland
[2]University of Zurich, Department of Geography, Soil Science and Biogeochemistry, Zurich, Switzerland

*Correspondence to*: Pranav Hirave (pranav.hirave@unibas.ch)

**Abstract.** Application of compound-specific isotope analysis (CSIA) in sediment fingerprinting source apportionment studies is becoming more frequent, as it can potentially provide robust land-use based source attribution of suspended sediments in a freshwater system. Isotopic tracers such as $\delta^{13}C$ values of vegetation-derived organic compounds are considered to be suitable for CSIA based fingerprinting method. However, a rigorous evaluation of tracer conservativeness in terms of the stability of isotopic signature during detachment and transport of soil during erosion process is essential for the suitability of the method. With the aim to identify potential fractionation and shifts in tracer signature during early degradation of organic matter in surface soils, we measured concentrations and $\delta^{13}C$ values of long-chain fatty acids and *n*-alkanes from fresh plant biomass (as vegetation is a direct source of these compounds to the soils), degraded organic horizon (O horizon) as well as mineral soil (A horizon) from various forest types with different humus forms (five sites). The bulk $\delta^{13}C$ values showed continuous $^{13}C$ enrichment through the degradation stages from fresh plant material to the O and A horizon, ranging between 3.5 and 5.6‰. Compound-specific $\delta^{13}C$ values showed a general $^{13}C$ enrichment for both, long-chain fatty acids (up to 5‰) as well as *n*-alkanes (up to 3.9 ‰) from fresh plant biomass to the O horizon overlying the A horizon. However, only slight or no further changes occurred from the O to the A horizon. We also compared compound-specific $\delta^{13}C$ values between two soil particle-size classes (< 2 mm and < 63 µm) from four sites and found no significant differences of tracer values between them, with even less fractionation for the long-chain *n*-alkanes within the soil particle fractions, which points to the conclusion that sampling and analysing bulk soil material might be valid for the isotopic tracer applications. We further conclude, that our results support the suitability of studied isotopic tracers as representative source soil signature in CSIA based sediment source attribution, as they demonstrated necessary stability in plant-soil system during organic matter degradation.

## 1 Introduction

Sediment input to water bodies by water erosion is one of the most prominent causes of freshwater ecosystem degradation. The detrimental effects of high sediment inputs to the water bodies include contamination of water with nutrients and particle-bound chemicals implying the degradation of aquatic habitat quality (Owens et al., 2005) as well as effects on infrastructure e.g., the economic lifespan of the water reservoir and its storage capacity (Haregeweyn et al., 2012; Wisser et al., 2013). Given these circumstances, methods to identify and quantify the sediment sources by various fingerprinting techniques has been in the focus of many scientific studies as well as international policy frameworks (Evrard et al., 2011; Gibbs, 2008; Glendell et al., 2018; Martínez-Carreras et al., 2010; Wallbrink et al., 1998).

The fingerprinting based on the compound-specific isotope analysis (CSIA) technique uses the stable isotope ratios of carbon and/or hydrogen of specific organic compounds (biomarkers) as tracers that are vegetation specific. Upon synthesis by higher terrestrial plants, these organic compounds including long-chain fatty acids and *n*-alkanes are continuously transferred into the soil through litter deposition and abrasion of cuticular waxes (Glaser, 2005; Jansen and Wiesenberg, 2017). The vegetation-specific isotopic signal variability, particularly in $\delta^{13}C$ values of these compound classes rising from biological (i.e., photosynthesis) and environmental factors (i.e., water stress) is also transferred into the soil. This allows us to discriminate the sediment sources by the fraction attributable to the major land-uses present in the catchment. Various organic compounds have been used as sediment tracers such as the carbon isotope ratios ($\delta^{13}C$ values) of individual long-chain fatty acids (Alewell et al., 2016; Blake et al., 2012; Brandt et al., 2016; Gibbs, 2008; Hancock and Revill, 2013; Mabit et al., 2018), long-chain *n*-alkanes (Chen et al., 2016; Cooper et al., 2015; Seki et al., 2010) and plant family-specific biomarkers (e.g., triterpenyl acetates produced by Asteraceae) (Lavrieux et al., 2019). Also the isotopic composition of the compounds is supported by the molecular proxies based on the concentration of these compounds such as the average chain length (ACL) values and carbon preference index (CPI) (Cooper et al., 2015).

The studies employing CSIA based sediment fingerprinting technique have mainly relied on the previously assessed suitability of these tracers for palaeoclimate and palaeovegetation reconstruction applications (Eglinton and Eglinton, 2008; Glaser and Zech, 2005). Huang et al., (1997) reported long-term degradation and diagenesis processes did not result in changes of carbon isotope composition in plant *n*-alkanes. Also it is believed that $\delta^{13}C$ signature of organic compounds does not change by volatilization, dilution, dispersion and sorption (Blessing et al., 2008 and references therein). However, large uncertainty is observed in the isotopic stability of these tracers in the plant-soil systems (Chikaraishi and Naraoka, 2006; Nguyen Tu et al., 2011). Some studies report several per mil enrichment in *n*-alkane $\delta^{13}C$ values of the soils compared to the dominant higher terrestrial plants (Ficken et al., 1998; Griepentrog et al., 2016; Nguyen Tu et al., 2004; Wiesenberg et al., 2004). Hence the effects of early degradation on these plant-derived chemical assemblages across the soil horizons, in terms of their carbon isotope ratios and molecular distributions need to be understood in greater detail. Specifically the degradation stage in the surface soils at which the tracer signature becomes stable and shows no further changes needs to be identified not only for a reliable identification of source signature for CSIA based sediment fingerprinting technique but also for an assessment of the relative stability of the isotopic signatures during detachment and transport of sediments. Our approach is to compare original C isotopic signature from fresh plant material with organic matter from the organic horizons and upper mineral soil horizon (e.g., increasing degradation stages along the organic horizons – top-soil profile). Also, it has been shown that within the soil particle-size fractions and density fractions the stability and composition of the tracer signal may change (Cayet and Lichtfouse, 2001; Griepentrog et al., 2016; Quénéa et al., 2006). In previous CSIA based fingerprinting studies, the silt-clay fraction (< 63 µm) or < 100 µm fraction has been chosen to extract the representative source signature (Blake et al., 2012; Cooper et al., 2015; Gibbs, 2008; Hancock and Revill, 2013). The bulk soil fraction (< 2 mm) has also been used (Alewell et al., 2016) and it is more common in soil OM source and turnover studies (Wiesenberg et al., 2004; Zocatelli et al., 2012).

We measured carbon isotopic composition and concentration of long-chain fatty acids and *n*-alkanes from fresh leaves, needles, mosses and degraded organic matter horizons and mineral soil (Oi-Oe-Oa-A horizons) from various forest types with different humus forms in order to (1) understand the effect of early degradation on the tracer signals employed in CSIA based sediment fingerprinting technique and (2) compare the two biomarker classes in terms of their $\delta^{13}C$ values and molecular distribution in the plant-soil system for potential suitability as representative source soil signature for

sediment fingerprinting. We also analysed the compound-specific $\delta^{13}C$ values from two soil particle-size fractions (< 2 mm and < 63 µm) to (3) compare the tracer signatures between them to understand the size dependent isotopic fractionation and its implication for sediment source attribution.

## 2 Materials and methods

### 2.1 Study area

For studying the degradation effect on the vegetation-specific tracers, five sampling sites were selected (Fig. 1); of which two are in the Southern Black Forest (Germany), two in the Lake Baldegg catchment (Switzerland) and one in the Upper Sûre Lake catchment (Luxembourg). The broadleaved forest site in the Southern Black Forest (site 1; BFbeech) was mainly stocked with beech (*Fagus sylvatica*), under which a moder (duff mull) type humus with clearly separable Oi and Oe sub-horizons but a diffused boundary between O and A horizon has developed. The second site in Southern Black Forest is a coniferous forest where the sole tree species is spruce (*Picea abies*, site 2a; BFspruce) and a thick cover of moss (*Sphagnum quinquefarium*, site 2b; BFmoss) is growing under very acidic conditions (pH ~3) characterised by the development of raw humus with an easily separable sequence of Oi-Oe-Oa and A horizons. One of the two sites in the Lake Baldegg catchment has mull type humus with a thin Oi layer on top of the A layer (Site 4; LBspruce). The mixed-forest site in the Upper Sûre Lake catchment has oak (*Quercus robur*) and spruce (*Picea abies*) cover with moder type humus (site 5; LXmixed). All sampling sites have acidic soils with pH values in the range of 3-4.2. As we are aware that a range of soil pH values might have been preferable, fully developed organic horizons are not found in slightly acidic or alkaline soils.

### 2.2 Sampling

Fresh leaves, needles and mosses were collected in autumn. Multiple leaves/needles (15-20) from different directions at ca. 3 m above ground were collected from an individual tree and mixed to form a sample of fresh material. The O horizon was sampled and sub-horizons within it were separated according to the humus form observed. Underlying A horizon was sampled to a maximum depth of 5 cm. Each sample was formed from a composite mixture of 3 sub-samples taken at a distance of ca. 1 m from each other.

### 2.3 Sample preparation and bulk isotope analysis

Collected samples were oven-dried at 40°C for 72 h. Soils were dry sieved to a particle size < 2 mm and macroscopic plant remains and stones were removed with tweezers. All the samples were roughly ground with a mortar and pestle. An aliquot of ground samples was finely powdered with a ball mill (Retsch MM400, Retsch GmbH, Haan, Germany) for 90 s at a frequency of 24 s $min^{-1}$ for analysis of bulk carbon concentration and isotopic ($\delta^{13}C$) composition. Powdered samples were weighed in tin capsules and subsequently stable carbon isotope ratios ($\delta^{13}C_{bulk}$) and C content were measured with an elemental analyser coupled to an isotope ratio mass spectrometer (EA/IRMS) (Sercon Integra2, Sercon Ltd., Crewe, UK). The carbon isotope values of samples are reported in a delta notation relative to the V-PDB standard. The instrumental standard deviation is lower than 0.1‰ for $\delta^{13}C$.

### 2.4 Particle size separation

The particle fraction of size < 63 µm was separated from four A horizon soil samples collected from various forest types (beech forest (site 1), coniferous forest (site 2a), coniferous forest with moss cover (site 3) and mixed forest (site 5)). Prior to the separation, soil aggregates were disrupted by ultrasonic dispersion using an ultrasonic probe (Branson 250 model, Branson Ultrasonics, Danbury, CT, USA). A 500 J $ml^{-1}$ ultrasonic energy was applied as described in (Schmidt et al., 1999). This energy is sufficient for total disruption of aggregates without redistribution and compositional changes in OM. After disruption, the fraction of size < 63 µm was separated by wet sieving and water was removed by sedimentation and centrifugation.

### 2.5 Total lipid extraction and compounds separation

Powdered samples were extracted using an accelerated solvent extraction system (ASE 350, Dionex Corp., Sunnyvale, CA, USA) with dichloromethane (DCM)/methanol (MeOH) (9:1, v/v) over three extraction cycles at 100°C. The total lipid extract (TLE) was volume reduced by vacuum evaporation (Rocket Evaporator, Genevac Ltd., Ipswich, UK). The TLE was subsequently separated into neutral, acidic and polar fraction by solid-phase extraction using aminopropyl-bonded silica gel as stationary phase (Jacob et al., 2005). The acidic fraction (including free fatty acids) was methylated at 60°C for 1 h using 1 mL of 12–14% boron trifluoride ($BF_3$) in MeOH and fatty acid methyl esters (FAMEs) were extracted with hexane and 0.1 M KCl. The aliphatic hydrocarbons containing long-chain *n*-alkanes were separated from neutral fraction obtained earlier using a column filled with an activated silica gel by eluting with *n*-heptane.

### 2.6 Biomarker analysis

Individual compounds in the FAMEs and aliphatic hydrocarbon fractions were identified using a TRACE GC Ultra gas chromatograph coupled with a DSQ II mass spectrometer (MS) (Thermo Scientific, Waltham, MA, USA). Quantification was performed using the same GC coupled with a flame ionization detector (FID) (Thermo Scientific, Waltham, MA, USA). The GC method was set identical during both measurements as described in Alewell et al., (2016).

### 2.7 Compound-specific isotope analysis (CSIA)

The $\delta^{13}C$ isotopic composition of individual FAMEs and *n*-alkanes was determined using a Trace 1310 GC with split/splitless injector and FID detector interfaced on-line via a GC-IsoLink II to a ConFlow IV and Delta V Plus isotope ratio mass spectrometer (Thermo Scientific, Bremen, Germany). A DB-5MS capillary column (50 × 0.2 mm i.d., 0.33 µm film thickness, J & W Scientific, Folsom, CA, USA) was used. The GC temperature program for *n*-alkanes was 70°C (held 4 min) to 320°C (held 35 min) at 5°C $min^{-1}$. The GC temperature program for FAMEs was 70°C (held 4 min) to 150°C at 20°C $min^{-1}$ and afterwards to 320°C (held 40 min) at 5°C $min^{-1}$. He (purity 5.0) was used as a carrier gas at a constant flow of 1 ml $min^{-1}$. $CO_2$ of known $\delta^{13}C$ isotopic composition used as reference gas was automatically introduced via a ConFlow IV into the isotope ratio mass spectrometer in a series of 5 pulses at the beginning and 4 pulses at the end of each analysis during every measurement. The system was externally calibrated with standard mixtures A6 for *n*-alkanes (*n*-$C_{16}$ to *n*-$C_{30}$) and F8-3 for FAMEs (even chain fatty acid methyl and ethyl esters from *n*-$C_{14:0}$ to *n*-$C_{20:0}$), both obtained from Arndt Schimmelmann, Indiana University, IN, USA with addition of isotopically characterised *n*-$C_{24:0}$, *n*-$C_{26:0}$, *n*-$C_{28:0}$ and *n*-$C_{30:0}$ FAMEs in F8-3 (Sigma-Aldrich, St. Louis, MO, USA). The carbon isotope values of samples are reported in delta notation relative to the standard V-PDB, averaging at least three replicate measurements. The reported

$\delta^{13}C$ values of FAMEs were corrected for the additional carbon atom introduced during methylation. The analytical uncertainty is lower than ±0.5‰.

## 3 Results and discussion

### 3.1 Concentration of biomarkers

The lipids from fresh leaves as well as O and A horizons showed even-over-odd predominance in long-chain fatty acids (> $n$-$C_{22:0}$) and odd-over-even predominance in $n$-alkanes (> $n$-$C_{25}$), a property usually ascribed to the higher-plant biomass derived lipids (Amblès et al., 1994a; Eglinton et al., 1962; Kolattukudy et al., 1976). Similar concentrations (relative abundance with regard to total organic matter) of both of the compound classes were observed between fresh plant biomass and degraded organic matter, with slight increase within O horizon (Fig. 2). We defined concentration in mineral soil also with regard to total soil organic matter and not to soil weight for its better comparability to the fresh plant biomass and O horizon. We found a strong increase in compound concentrations in the mineral soils collected from various sites (Fig. 2), except site $BF_{moss}$. The strong increase in the concentrations from O horizon to mineral soil (A horizon) indicates a preferential preservation of long-chain fatty acids and $n$-alkanes compared to soil bulk organic matter (Fig. 2). In general, long-chain fatty acids were relatively more abundant than $n$-alkanes in the plant–soil system, similar to previously reported findings (van Bergen et al., 1998; Chikaraishi and Naraoka, 2006; Naafs et al., 2004).

Interspecies variability was recorded by the molecular distribution patterns of the compounds in fresh plant biomass (Fig. S1a and b). The coniferous trees had relatively high concentrations of $n$-$C_{29}$ alkane and $n$-$C_{24:0}$ fatty acid. Fresh leaves of broadleaved trees showed relatively high concentrations of $n$-$C_{27}$ alkane and $n$-$C_{28:0}$ fatty acid. The *sphagnum* species were characterised by high $n$-$C_{23}$ or $n$-$C_{21}$ alkane, differing from conifers and broadleaved trees, also consistent with previously reported distribution pattern (Bush and McInerney, 2013; Maffei et al., 2004).

However, molecular distribution patterns observed in fresh plant biomass were not necessarily transferred to the O and A horizons (Fig. S1a and b). This was shown by the changes in average chain length (ACL) values of both compound classes (Fig. 3). Relatively higher ACL values of $n$-alkanes in the A horizon compared to overlying horizons from some of the sites might have been the result of preferential degradation of shorter chained homologues (< $n$-$C_{27}$) and preservation of longer chained homologues from $n$-$C_{27}$ and higher (van Bergen et al., 1998; Lichtfouse et al., 1998; Marseille et al., 1999). In case of long-chain fatty acids, changes in the molecular distribution patterns (Fig. S1a) and decrease in the ACL values (Fig. 3) from fresh matter to the A horizon was observed at most sites. This might be explained by the production of long-chain fatty acids (> $n$-$C_{20:0}$) in O and A horizon from the oxidation of other components of vegetation-derived lipids such as $n$-alkanes and $n$-alkanols (Amblès et al., 1994a, 1994b; Marseille et al., 1999) and also a contribution of shorter chained homologues from microorganisms (Chikaraishi and Naraoka, 2006; Lichtfouse et al., 1995).

### 3.2 Stable carbon isotopic composition ($\delta^{13}C$ values)

The bulk stable carbon isotope values were in between -32.4‰ and -26.1‰, similar to the typical $\delta^{13}C$ values of $C_3$ plant-soil system (Diefendorf et al., 2010). During degradation, the carbon isotopic composition generally showed consecutive enrichment in $\delta^{13}C$ values from fresh plant biomass (leaves, needles and mosses) via organic horizons to the mineral soil.

The $\delta^{13}C$ enrichment from fresh plant biomass to the A horizon varied between 3.5‰ ($BF_{beech}$) and 5.6‰ ($BF_{moss}$) (Fig. 4). Selective degradation of bulk organic matter with depleted carbon isotopic signature has been observed previously, but with less significant changes probably related to short timescales (Nguyen Tu et al., 2004). Our findings report consistent degradation (with *p*-value = 0.72 denoting similar extent of degradation across studied forest systems) as well as consecutive enrichment in $\delta^{13}C$ values through degradation stages (*p*-value $< 0.001$ denoting significantly different $\delta^{13}C$ values). Hence the cautious use of bulk carbon isotope values for using it as source soil signatures is necessary.

Compared to bulk isotopic composition of fresh plant biomass, compound-specific isotope values of long-chain fatty acids and *n*-alkanes were generally depleted by -3‰ and -3.7‰ respectively (Fig. 4 and 5), in agreement with previous studies (Collister et al., 1994; Huang et al., 1997; Nguyen Tu et al., 2011). The compound-specific $\delta^{13}C$ values of long-chain fatty acids ($n$-$C_{24:0}$ to $n$-$C_{30:0}$) and *n*-alkanes ($n$-$C_{25}$ to $n$-$C_{31}$) showed general enrichment in $\delta^{13}C$ values from fresh plant biomass to the O horizon overlying the mineral soil (A horizon), however only slight or no further changes to the A horizon (Fig. 5a-h, Table 2). Both the compound classes showed similar trends of enrichment at each site. Exceptionally, some *n*-alkane homologues showed a strong enrichment between O and A horizon, e.g., $BF_{moss}$ ($n$-$C_{25,27}$), $LB_{spruce*}$ ($n$-$C_{29}$) and $LX_{mixed}$ ($n$-$C_{27,29}$) sites. Fresh moss material exhibited the most depleted $\delta^{13}C$ values for both the compound classes, -37.7‰ ($n$-$C_{26:0}$ FA) and -40.4‰ ($n$-$C_{25}$ alkane) of *Sphagnum quinquefarium* ($BF_{moss}$) and -39.4‰ ($n$-$C_{26:0}$ FA) and -37‰ ($n$-$C_{27}$ alkane) of *Thuidium tamariscinum* ($LB_{spruce*}$), similar to previously reported *sphagnum n*-alkane isotope values (Brader et al., 2010). Sites 2a ($BF_{spruce}$) and 2b ($BF_{moss}$) both are located in a spruce forest with thick moss cover, hence the strong isotopic enrichment in $n$-$C_{25}$ and $n$-$C_{27}$ alkane (3.9‰ and 2.4‰ respectively) observed from O horizon to mineral soil (A horizon) at $BF_{moss}$ might have caused by significantly higher long-term input from spruce (with comparatively enriched $\delta^{13}C$ value) compared to the fresh and degraded moss to the mineral soil. Also site $LX_{mixed}$ has a vegetation cover of two distinct species (table 2). The difference in isotopic composition of fresh plant biomass of these two species along with its seasonal variability might be a cause of changes in $\delta^{13}C$ values during degradation. Also, isotopically distinct microbial input of long-chain compounds can not be excluded (Chikaraishi and Naraoka, 2006; Marseille et al., 1999).

At sites where single dominant vegetation is present, we observed enrichment up to 5‰ in long-chain fatty acids and up to 3.2‰ in long-chain *n*-alkanes from fresh plant biomass to A horizon, similar to the previous findings (Chikaraishi and Naraoka, 2006; Griepentrog et al., 2015, 2016). It was found that most of the degradation effect on isotopic composition took place up to the O horizon and slight or no further changes were observed during transfer to the A horizon from the overlying O horizon. Table 2 summarizes this finding where we compared the changes in compound-specific isotope values with the level of significance. We calculated the magnitude of enrichment/depletion of $\delta^{13}C$ values from mineral soil horizon with respect to fresh biomass ($\Delta_1$), and also with respect to the overlying O horizon ($\Delta_2$) as follows:

$$\Delta_1 = \delta^{13}C_{(\text{A horizon})} - \delta^{13}C_{(\text{fresh plant biomass})}$$

$$\Delta_2 = \delta^{13}C_{(\text{A horizon})} - \delta^{13}C_{(\text{overlying O horizon})}$$

For the calculation of $\Delta_2$, we excluded the $\delta^{13}C$ values of fresh moss to obtain $\delta^{13}C_{\text{(avg. fresh biomass)}}$ value. This was done to eliminate the overestimation of the magnitude of isotopic change, as mosses had considerably depleted isotope values compared to other fresh materials (Fig. 5).

The bulk $\delta^{13}C$ values showed continuous enrichment from fresh plant material through the degradation stages to the A horizon and did not exhibit the stability needed for the use as isotope tracer. Furthermore, fresh plant biomass generally exhibited a wider range of compound-specific $\delta^{13}C$ values (7.5‰ for $n$-$C_{25}$ alkane and 3.9‰ for $n$-$C_{26:0}$ fatty acid) than bulk isotope values (1.6‰) across all the sites. Hence in terms of the stability during early degradation in soils and interspecies variability, our results confirm that the compound-specific isotope analysis is more useful technique for sediment source attribution and the analysis should not rely on bulk isotope values.

### 3.3 Compound-specific isotopic composition of soil particle-size fractions

We compared the compound-specific $\delta^{13}C$ values of long-chain fatty acids and $n$-alkanes from the bulk soil (< 2 mm) to the fine silt plus clay fraction (< 63 µm) (Fig. 6). None of the fatty acids showed a significant difference in their $\delta^{13}C$ values between the two soil particle size fractions. Only for $LB_{spruce*}$ site, $\delta^{13}C$ values of long-chain fatty acids were considerably (up to 2.1‰), but not significantly different among the studied fractions. Similar results were obtained for long-chain $n$-alkanes, where none of the $n$-alkanes from studied sites differed significantly between the different particle size fractions, with difference of up to 1‰ (Fig. 6). Cayet and Lichtfouse (2001) reported preferential input of long-chain $n$-alkanes in soils via larger (200-2000 µm) particle-size fraction. However in terms of isotopic composition of soils, Quénéa et al., (2006) reported less than 1‰ difference in long-chain $n$-alkanes $\delta^{13}C$ among the soil size fractions. In accordance with previous findings on $n$-alkanes, our results simultaneously studying long-chain fatty acids as well as $n$-alkanes, confirm that the compound-specific carbon isotope values of both compound classes were not significantly different between the two particle-sizes. Long-chain $n$-alkanes were found to be even less prone to the isotopic fractionation within the soil particle-size fractions. These results indicate that the analysis of bulk soil material instead of the fine soil fraction (< 63 µm) might be suitable for sediment source attribution from soils to freshwater systems, which might not only increase work and cost effectiveness of the method but also reduce potential error sources due to particle size fractionation effects. In general, above results further strengthen the necessary requirement of conservativeness of tracer signature during erosion processes like detachment, dispersion and sediment transport, for CSIA based sediment source apportionment.

## 4 Conclusions

We tested the potential stability of compound-specific carbon isotope values of long chain fatty acid and $n$-alkanes during early degradation in the soil; usually employed as tracers in compound-specific isotope analysis based sediment source fingerprinting technique. We assumed that a potential stability of a tracer signature during degradation stages from fresh plant material to the mineral soil might indicate its suitability for detachment and transport processes during soil erosion. We determined a clear enrichment in $\delta^{13}C$ values for both compound classes from aboveground plant biomass to the O horizon overlaying the mineral soil (A horizon), however only slight or no further changes were observed between the O and A horizon. The latter was in clear contrast to bulk $\delta^{13}C$ values, which continued to become enriched in $^{13}C$ through all degradation stages from fresh plant biomass to soil mineral horizon.

We conclude that this finding emphasizes the suitability of compound-specific isotope analysis compared to the bulk $\delta^{13}C$ analysis for sediment fingerprinting source attribution using stable carbon isotope. The finding was consistent through the various forest types with different vegetation and humus forms studied. The degree of variability in

compound-specific carbon isotopic signatures in the plant-soil system was mostly similar between both compound classes (long-chain fatty acids and *n*-alkanes). Comparing the tracer signatures between bulk soil material and the fine soil particle-size fraction (< 63 µm), the long-chain *n*-alkanes were found to be less prone to the particle-size dependent isotopic fractionation than fatty acids, however none of the compounds showed significant differences between studied particle sizes. Hence we suggest that our results indicate that the bulk soil (< 2 mm) can be safely used to determine tracer signatures in CSIA based sediment source fingerprinting, irrespective of particle-size of sediments collected. The 10-fold higher concentrations of the long-chain fatty acids compared to *n*-alkanes in surface soils suggests a better analytical certainty in extraction and measurements of fatty acids in soils and sediments with low lipid content, but the joint investigation of fatty acids and *n*-alkanes would add an additional component in CSIA based sediment source fingerprinting and thus more robust results can be expected, if analytically reliable quantity of *n*-alkanes is extracted from samples.

**Data availability**

Data from isotopic and molecular measurements is available at https://doi.org/10.5281/zenodo.3228446.

**Acknowledgements**

We thank Dr. Marlène Lavrieux for her support in sampling and Dr. Thomas Kuhn, from University of Basel for his assistance in isotope measurements.

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

**Table 1: Main characteristics of the investigated sites.**

| Site area | Forest type | Site/sub site No. | Name | Dominant vegetation cover | Humus form | Soil type | MAT (ºC) | MAAP (mm) | Altitude (m.a.s.l.) |
|---|---|---|---|---|---|---|---|---|---|
| Southern Black Forest | Beech forest | Site 1 | $BF_{beech}$ | *Fagus sylvatica* (beech) | Moder | Stagnosol | 8.8 | 1155 | 535 |
| | Coniferous forest | Site 2a | $BF_{spruce}$ | *Picea abies* (spruce) | Raw humus | Podzol | 3.9 | 1800 | 965 |
| | | Site 2b | $BF_{moss}$ | *Sphagnum quinquefarium* (moss) | Raw humus | Podzol | 3.9 | 1800 | 965 |
| Lake Baldegg catchment | Coniferous forest | Site 3 | $LB_{spruce*}$ | *Picea abies* (spruce), *Thuidium tamariscinum* (moss) | Raw humus | Cambisol | 8.4 | 1100 | 608 |
| | Coniferous forest | Site 4 | $LB_{spruce}$ | *Picea abies* (spruce) | Mull | Cambisol | 8.4 | 1100 | 573 |
| Upper Sûre Lake catchment | Mixed forest | Site 5 | $LX_{mixed}$ | *Picea abies* (spruce), *Quercus robur* (oak) | Moder | Cambisol | 9.2 | 970 | 429 |

**Table 2: Range of compound-specific $\delta^{13}C$ values between the A horizon and fresh plant biomass average ($\Delta_1$), and between the A horizon and overlying O horizon ($\Delta_2$) (Significance levels derived from an Anova/Tukey's HSD Test for the difference in $\delta^{13}C$ values, given in parentheses).**

| Site Name | Change in $\delta^{13}C$ value (‰) | Fatty acids | | | | *n*-Alkanes | | | |
|---|---|---|---|---|---|---|---|---|---|
| | | *n*-$C_{24:0}$ | *n*-$C_{26:0}$ | *n*-$C_{28:0}$ | *n*-$C_{30:0}$ | *n*-$C_{25}$ | *n*-$C_{27}$ | *n*-$C_{29}$ | *n*-$C_{31}$ |
| $BF_{beech}$ | $\Delta_1$ | 4.0 (***) | 3.6 (**) | 3.0 (**) | 1.7 (n.s.) | 3.1 (**) | 3.2 (**) | 2.9 (**) | - |
| | $\Delta_2$ | 0.9 (n.s.) | 0.4 (n.s.) | 0.8 (n.s.) | -1.9 (n.s.) | 0.2 (n.s.) | -0.1 (n.s.) | -0.9 (n.s.) | - |
| $BF_{spruce}$ | $\Delta_1$ | 5.0 (*) | 1.8 (n.s.) | 0.2 (n.s.) | 0.2 (n.s.) | 0.9 (n.s.) | 0.6 (n.s.) | 1.1 (n.s.) | - |
| | $\Delta_2$ | 0.0 (n.s.) | -0.7 (n.s.) | -0.3 (n.s.) | -0.2 (n.s.) | -0.3 (n.s.) | 0.8 (n.s.) | 0.5 (n.s.) | - |
| $BF_{moss}$ | $\Delta_1$ | 5.2 (*) | 4.6 (n.s.) | 1.3 (n.s.) | -0.3 (n.s.) | 7.8 (**) | 5.3 (*) | 1.7 (n.s.) | 1.7 (n.s.) |
| | $\Delta_2$ | 0.3 (n.s.) | 1.1 (n.s.) | 0.7 (n.s.) | -0.3 (n.s.) | 3.9 (n.s.) | 2.4 (n.s.) | 1.4 (n.s.) | 0.6 (n.s.) |
| $LB_{spruce*}$ | $\Delta_1$ | 3.1 (n.s.) | 0.6 (n.s.) | -0.5 (n.s.) | -1.5 (n.s.) | 2.3 (n.s.) | 1.6 (n.s.) | 0.5 (n.s.) | - |
| | $\Delta_2$ | 0.8 (n.s.) | 1.0 (n.s.) | 0.3 (n.s.) | 0.2 (n.s.) | 0.5 (n.s.) | -0.3 (n.s.) | -1.3 (n.s.) | - |
| $LB_{spruce}$ | $\Delta_1$ | 3.8 (n.s.) | 1.6 (n.s.) | 0.0 (n.s.) | -0.4 (n.s.) | - | - | - | - |
| | $\Delta_2$ | 2.0 (n.s.) | 1.9 (n.s.) | -0.7 (n.s.) | -1.1 (n.s.) | 1.3 (n.s.) | 1.0 (n.s.) | 0.4 (n.s.) | -1.0 (n.s.) |
| $LX_{mixed}$ | $\Delta_1$ | -0.5 (n.s.) | -0.7 (n.s.) | -0.8 (n.s.) | -0.7 (n.s.) | 0.2 (n.s.) | 1.0 (n.s.) | 1.1 (n.s.) | -0.9 (n.s.) |
| | $\Delta_2$ | 1.5 (n.s.) | 0.8 (n.s.) | 0.9 (n.s.) | 0.6 (n.s.) | 0.8 (n.s.) | 2.2 (*) | 2.5 (**) | 1.4 (n.s.) |

Significance codes: non-significant ($p > 0.05$) 'n.s.'; $p \leq 0.05$ '*'; $p < 0.01$ '**'; $p < 0.001$ '***'.

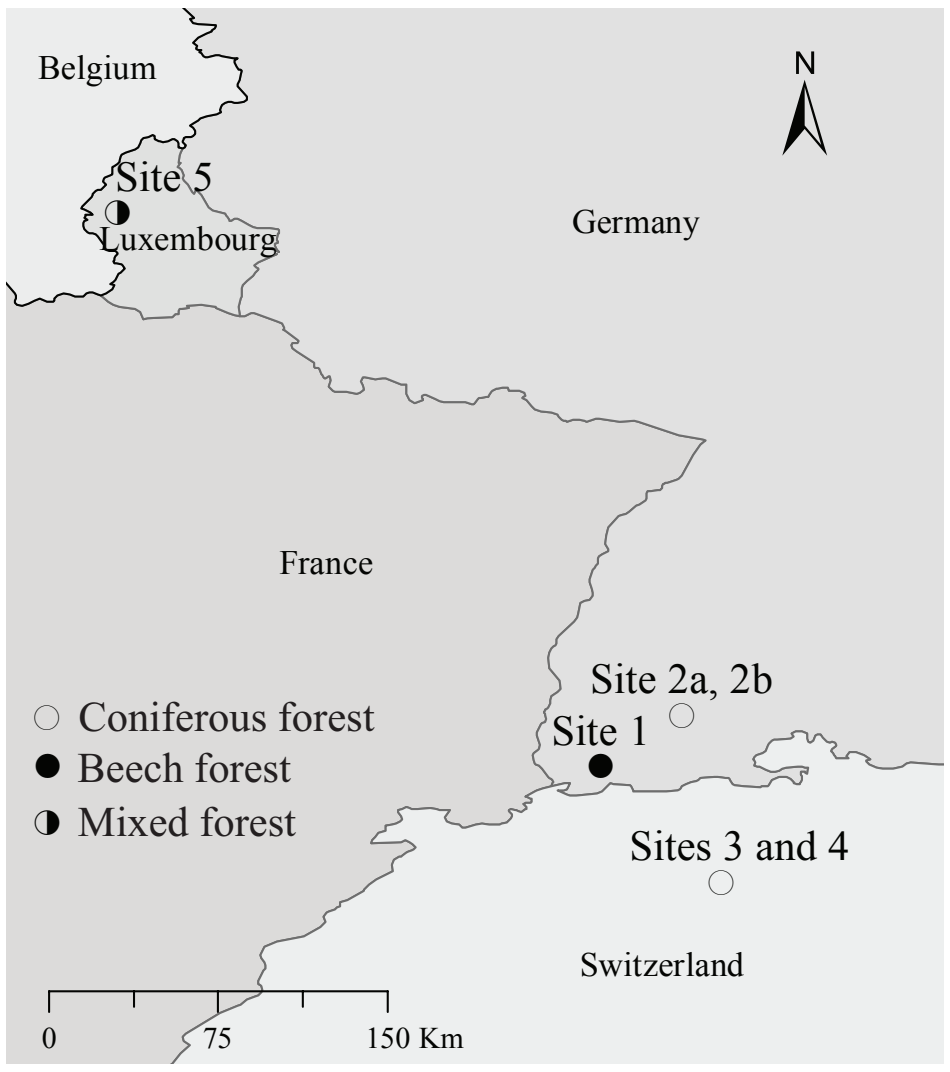

**Figure 1: Map of all sampling sites.**

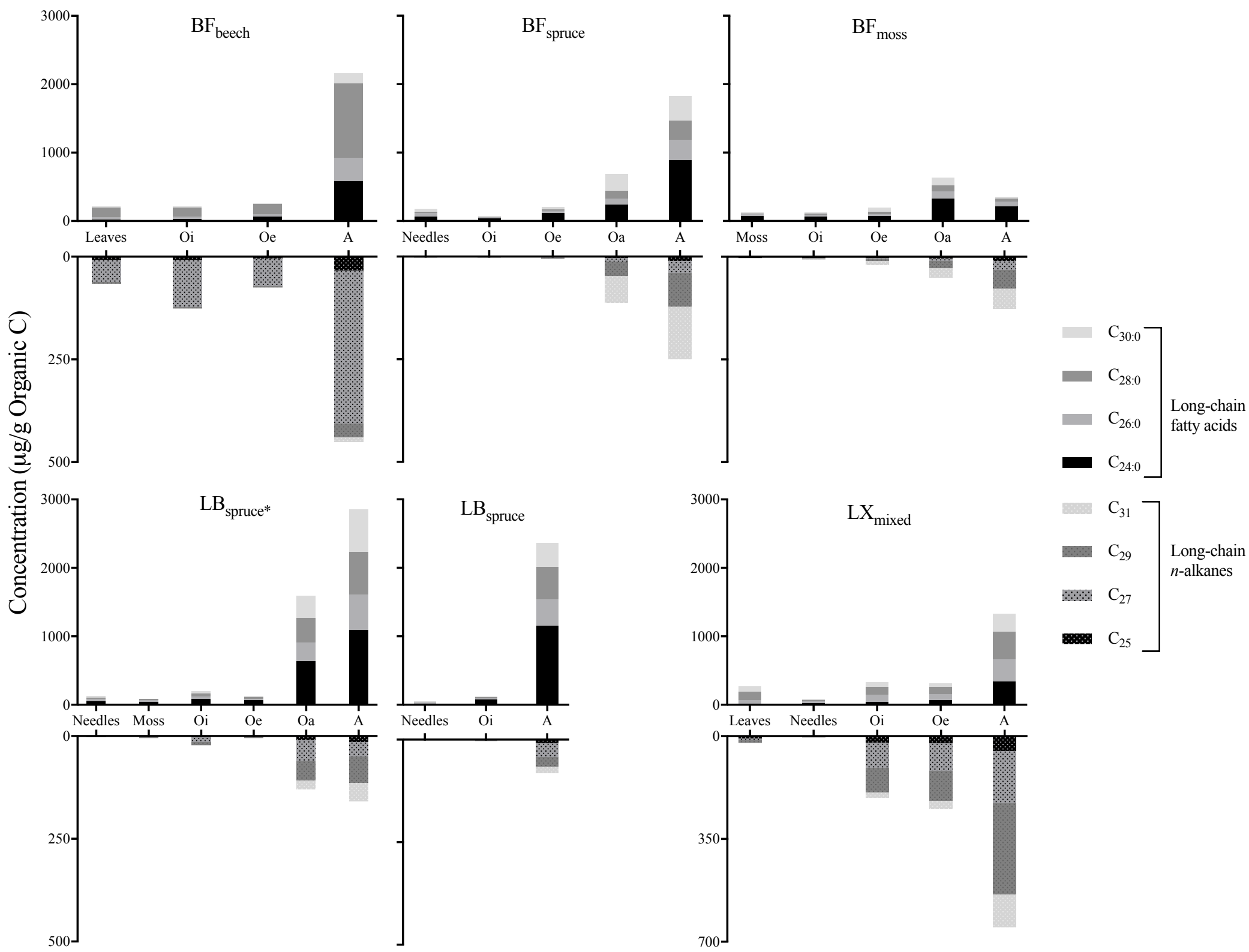

**Figure 2: Concentration of long-chain fatty acids and *n*-alkanes in fresh plant biomass and underlying horizons from different sites (note the different scales of fatty acid and *n*-alkane concentration).**

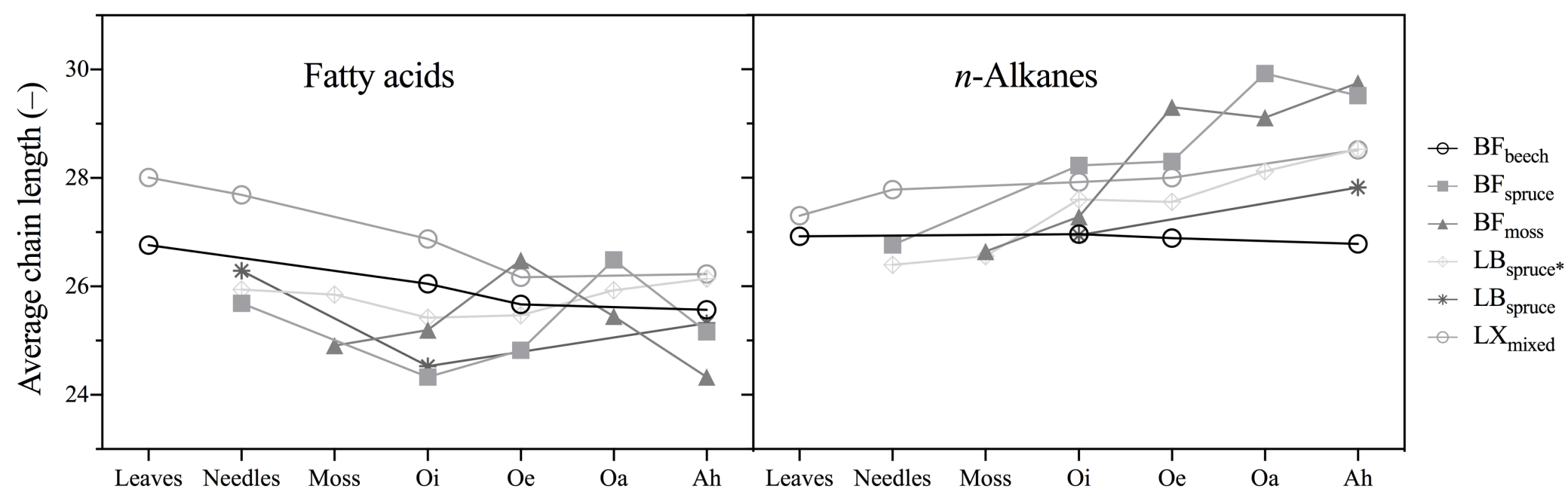

**Figure 3: Average chain length (ACL) of fatty acids ($ACL_{22-32}$) and *n*-alkanes ($ACL_{21-33}$).**

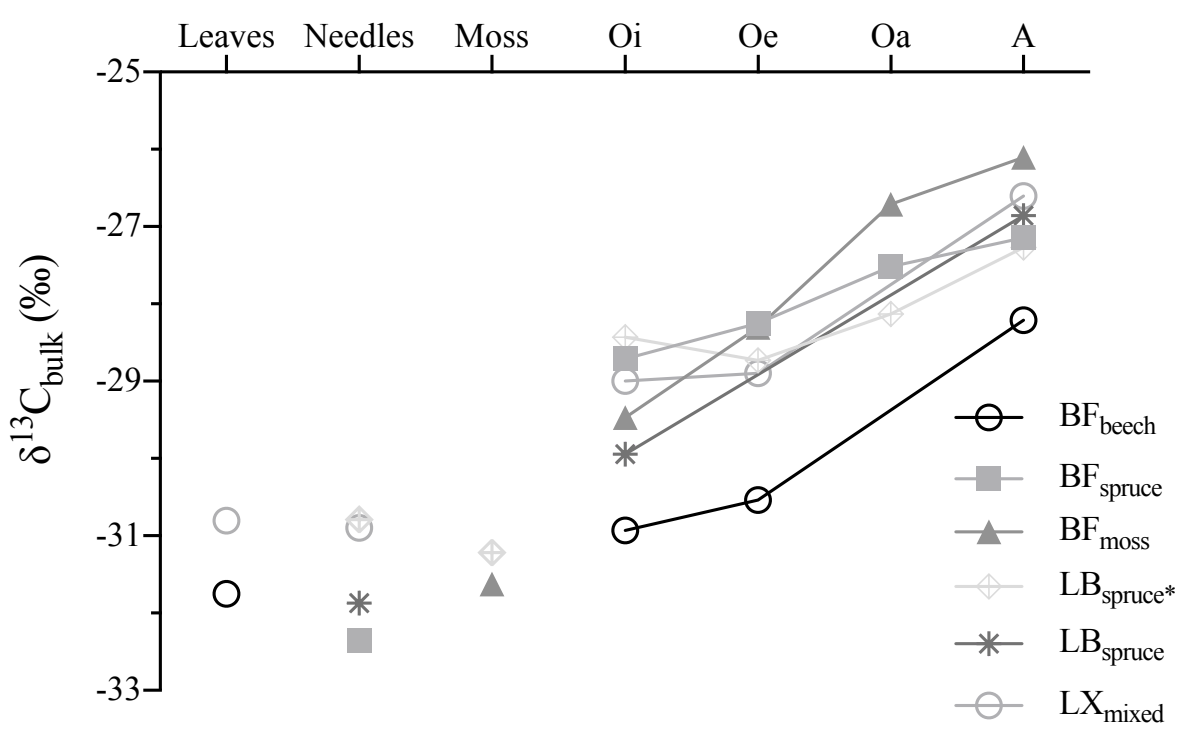

**Figure 4: Bulk $\delta^{13}C$ values (‰) of fresh plant biomass (leaves, needles & mosses) and underlying horizons.**

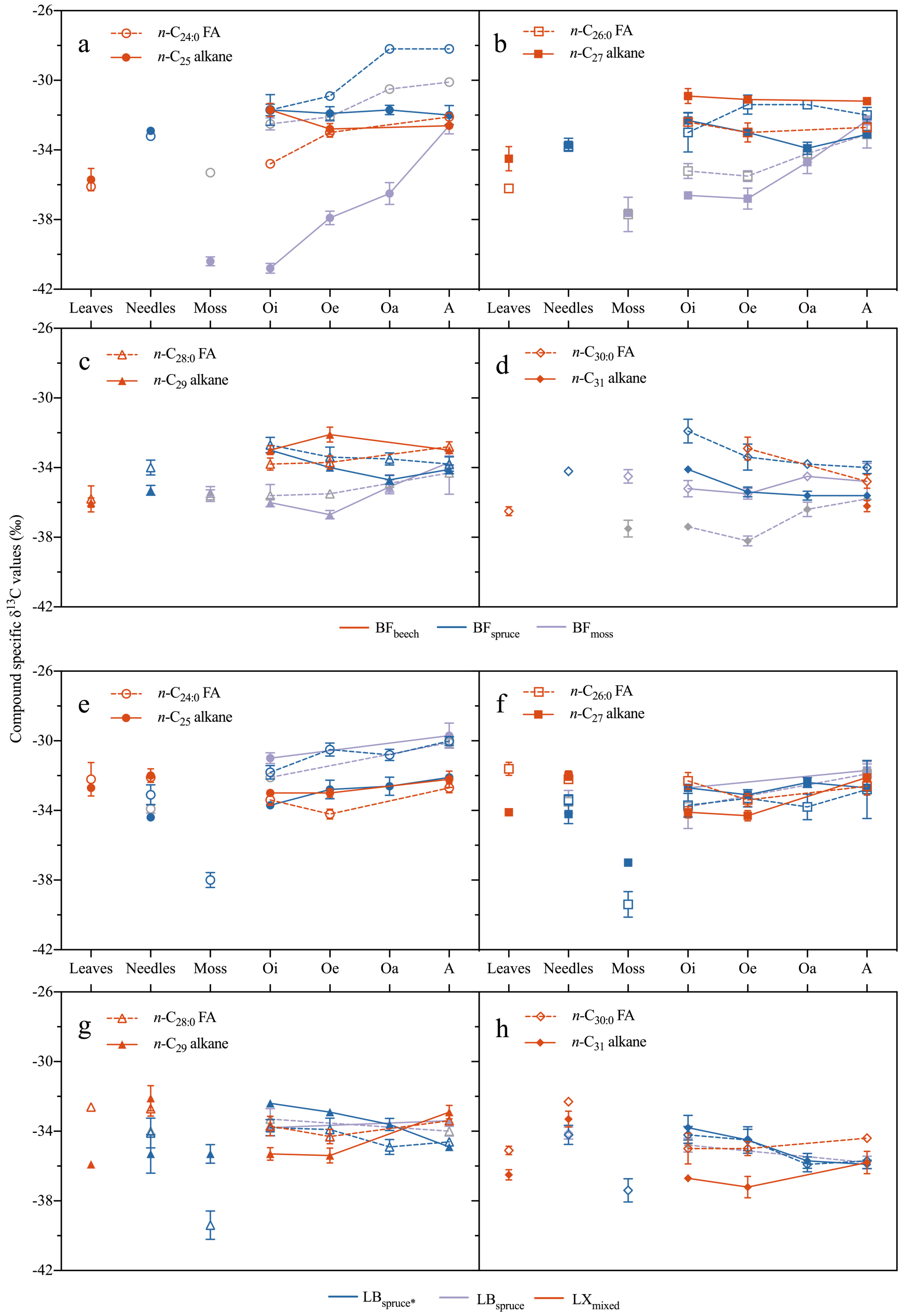

**Figure 5: Compound-specific $\delta^{13}C$ values of long-chain fatty acids ($n$-$C_{24:0}$ to $n$-$C_{30:0}$) and long-chain $n$-alkanes ($n$-$C_{25}$ to $n$-$C_{31}$) in fresh plant biomass (leaves, needles & mosses) and across the underlying horizons (mean ± measurement error of 3 replicates).**

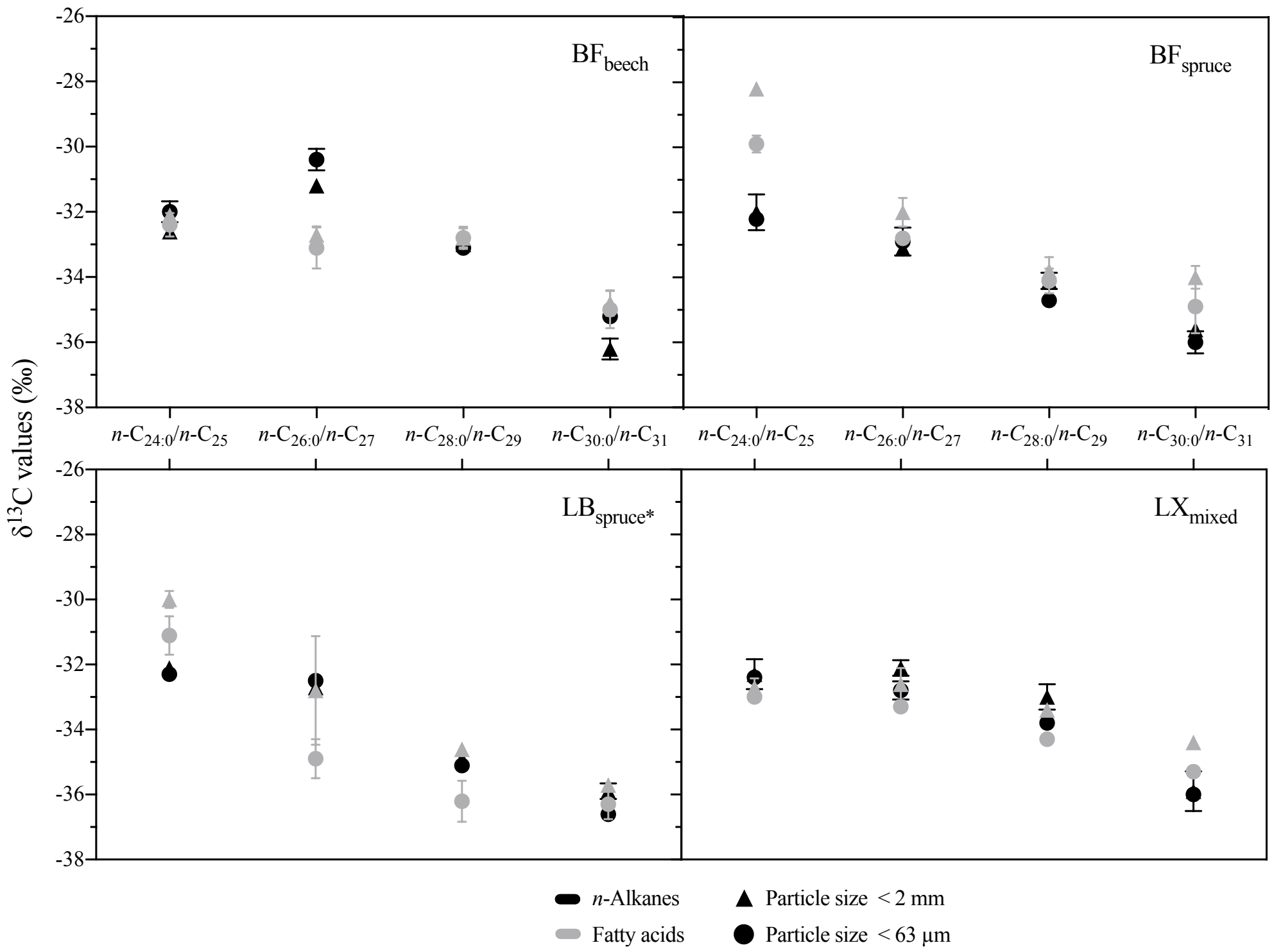

**Figure 6: Comparison of the $\delta^{13}C$ values from two particle-size fractions in soil from four sites (mean ± measurement error of 3 replicates).**