# Peer review of "Understanding the effects of early degradation on isotopic tracers: implications for sediment source attribution using compound-specific isotope analysis (CSIA)"

_Biogeosciences, 2019_

## Referee Comment (RC1) · Anonymous Referee #1 · 15 Jul 2019

Review on the manuscript 'Understanding the effects of early degradation on isotopic tracers: implications for sediment source attribution using compound specific isotope analysis (CSIA)' by Hirave et al.,

The manuscript deals with the robustness of CSIA values of long chain alkanes and fatty acids as sediment source finger printing during early degradation in the soil. Their results show that bulk d13C values show enrichment from plant biomass to the soil mineral horizon, whereas CSIA of fatty acids and alkane showed enrichment from plant biomass until the O horizon overlaying the mineral soil. Also their results show that

bulk soil (<2 mm) can be safely used to determine tracer signatures in CSIA- based sediment source finger printing irrespective of particle size of sediments.

Overall the manuscript is well written. But certain specific comments may be addressed before publication.

Abstract:

Page 1 lines 8-10: The application of CSIA is not restricted only to freshwater systems, as many studies from marine environment (For eg., Canuel et al., 1997; Limnol Oceano, 42, 1570) also show its importance.

Introduction:

Page 2 line 16: Huang et al. (1997) Punctuation- please remove comma here and many other places Page 2 line 28: I feel it should be 'fresh plant material with organic matter to the organic horizons and upper mineral soil horizon'

Carbon preference Index of alkanes can also be given in addition to average chain length.

2. Materials and methods

Page 3, line 13: It is better to define Oi-Oe-Oa horizons to the readers at the first time

Line 15: 'moder' type spelling correction

2.4 Page 4 line 7: Please define OM at the first usage Line 5: please remove the bracket for reference Schmidt et al., 1999

Page 5 line 1: please elaborate or give reference how the correction for d13C values was applied for additional carbon atom introduced during methylation.

Section 3.2 summarizes that bulk d13C values should not be relied for sediment source attribution compared to compound specific isotope values. However the issues/reasons (other than enrichment) while considering bulk isotope values are not

discussed in detail.

The manuscript is written well, but the novelty and importance of present study need to be highlighted in the manuscript.

Citation of references in the text is given alphabetical order. Normally it is cited in chronological order (ascending). Anyway please follow the journal format and modify accordingly.

The following reference may be cited if found applicable. They also emphasize the usage of d13C of >C 20 fatty acids for source apportionment compared to shot chain fatty acids.

Upadhayay et al., (2017) Methodological perspectives on the application of compound-specific stable isotope fingerprinting for sediment source apportionment; J Soils Sediments (2017) 17:1537–1553; DOI 10.1007/s11368-017-1706-4

The manuscript can be accepted after minor revision.

---

## Author Comment (AC1) · 30 Jul 2019

We thank the referee for his/her valuable suggestions, which will further improve the quality of our manuscript. At this stage, we would like to give a short reply to the referees' main comments. We will post a final response with answers to all comments along with the revision of our manuscript after having seen the comments of all referees.

Please also note the supplement to this comment:

https://www.biogeosciences-discuss.net/bg-2019-205/bg-2019-205-AC1-supplement.pdf

**Supplement:**

We thank the referee for his/her valuable suggestions, which will further improve the quality of our manuscript. At this stage, we would like to give a short reply to the reviewers' main comments. We will post a final response with answers to all comments along with the revision of our manuscript after having seen the comments of all referees.

Referee comments are in blue.

Abstract:
Page 1 lines 8-10: The application of CSIA is not restricted only to freshwater systems, as many studies from marine environment (For eg., Canuel et al., 1997; Limnol
Oceano, 42, 1570) also show its importance

Yes, for sure, we did not meant to imply this. In fact, application of CSIA was earlier and is already more advanced in marine systems. We will add a sentence referring to marine systems and knowledge in the revised manuscript.

Introduction:
Page 2 line 16: Huang et al. (1997) Punctuation- please remove comma here and many other places
Page 2 line 28: I feel it should be 'fresh plant material with organic matter to the organic horizons and upper mineral soil horizon'

We will correct the formatting and also restructure the sentence.

Carbon preference Index of alkanes can also be given in addition to average chain length.

CPI values were in support of our finding of general odd over even predominance in long-chain *n*-alkanes through all the degradation stages, which we have mentioned in results and discussion on Page 5 line 5. We will add a bit more on CPI values to enhance this result in the manuscript.

2. Materials and methods
Page 3, line 13: It is better to define Oi-Oe-Oa horizons to the readers at the first time
Line 15: 'moder' type spelling correction
2.4 Page 4 line 7: Please define OM at the first usage Line 5: please remove the bracket for reference Schmidt et al., 1999

Thank you for pointing out improvements here, we will add the relevant information in materials and methods section.

Section 3.2 summarizes that bulk d13C values should not be relied for sediment source attribution compared to compound specific isotope values. However the is- sues/reasons (other than enrichment) while considering bulk isotope values are not discussed in detail.

Our aim of this study is to test if isotopic signatures of tracers used for sediment source attribution are stable which requires non-fractionation during detachment and transport. Our assumption is that if they are stable during degradation, they will also be stable during detachment and transport. Thus, the continuous enrichment in bulk $\delta^{13}$C values through the degradation stages is *the* reason of concluding that they are not suitable as sediment tracers. We do not know of any other major reason to reject bulk isotope analysis for sediment source attribution. Sorry, if this was not clear enough, we will have a critical look at the formulation of our aims and assumptions. We would like to add, that source apportionment with differences in $C_3$- and $C_4$- vegetation sources has been done successfully, because differences in bulk isotopic signals of sources are bigger, thus small deviations by fractionation might not substantially alter the results.

The manuscript is written well, but the novelty and importance of present study need to be highlighted in the manuscript.

Thank you and also for pointing out, that we should highlight our results more. We will add more information on the novelty and uniqueness of the study; especially as the results are from different forest types. We will also emphasize the importance of our study for research questions beyond sediment source attribution.

Citation of references in the text is given alphabetical order. Normally it is cited in chronological order (ascending). Anyway please follow the journal format and modify accordingly.

Publication house states that the in-text citations can be written in alphabetical or chronological order, but as generally the order is chronological, we will change it as suggested.

We are also happy to add Upadhayay et al. 2017 mentioning it in an appropriate section in the manuscript.

---

## Referee Comment (RC2) · Anonymous Referee #2 · 6 Nov 2019

Review comments on: "Understanding the effects of early degradation on isotopic tracers: implications for sediment attribution using compound-specific isotope analysis (CSIA) By Hirave et al.

The authors have made an attempt to fingerprint organic matter source using compound-specific isotope ratio measurement of organic matter from forest soil. They have tried to understand the degradation of fresh organic matter and change in carbon isotope ratios of n-alkane and long-chain fatty acids in different forest soil. The manuscript has reasonably well written. However, the authors may kindy address the

following issues for better clarity.

1. Introduction: Authors have started this section by introducing the causes of fresh ecosystem degradation as result of nutrient input along with detrital riverine/ aeolian input. However, they ended up with the determination of organic matter sources and extent of degradation in a soil horizon. The connectivity between organic matter source and concerned with freshwater ecosystems has not been addressed. This aspect needs to be elaborated in the discussion or modify the introductory part for better co-herence.

2. Methodology:

A. Why inorganic carbon was not removed from the soil prior to bulk carbon isotope ratio measurement? B. What is the recovery % of short and long-chain compounds in the entire extraction process? C. How the methylation correction was applied for carbon isotope ratio measurement? Which formula was used and how the alcohol isotope ratio was measured?

3. Result: A. The best way to show the result is against the respective soil profiles. The different zone identified in the different soil profiles can also be marked for clarity. B. What is LB pruce*? C. At many places, you are writing d13C enrichment or depletion!! Please write an 'increase in ïAď13C or 'enrichment of 13C'. D. Why moss values are neglected? Why the significantly low ïAď13C values? 4. Discussion: A. Why the bulk ïAď13C and is showing 13C enrichment? Compared to the nC28 and nC29 B. Is the true for other long-chain fatty acids and alkanes as well?

---

## Author Comment (AC2) · 3 Dec 2019

We thank the referee for his/her thorough review and valuable comments. We considered them one by one with great care and address them here with the hope for better clarity.
Referee comments are in blue.

1. Introduction:

Authors have started this section by introducing the causes of fresh ecosystem degradation as result of nutrient input along with detrital riverine/ aeolian input. However, they ended up with the determination of organic matter sources and extent of degradation in a soil horizon. The connectivity between organic matter source and concerned with freshwater ecosystems has not been addressed. This aspect needs to be elaborated in the discussion or modify the introductory part for better coherence.

Our aim of the study was to understand the implications of early degradation of organic material on isotopic tracer signatures in soils. These results we would then use to infer on interpretation of the sediment source attribution method. Hence the introduction is written in two parts; first, discussing the importance of sediment source attribution and the common tracers used in the technique. The other part addresses the issues of tracer stability during early degradation in soils, specifically the stability of long-chain fatty acid and *n*-alkanes after production and detachment, in terms of their carbon isotopic signature.

However, the reviewer raises a valid point and will considerably improve the coherence of the text, with less emphasis on the sediment source attribution and more focus on the stability of isotopic tracer signature. To address the connectivity between the stability of organic matter source signature and sediment tracing using CSIA in a freshwater ecosystem, we will include connecting sentences in the introduction section emphasizing the importance of understanding the stability of organic matter isotope signature in connection to CSIA based fingerprinting method.

2. Methodology:

A. Why inorganic carbon was not removed from the soil prior to bulk carbon isotope ratio measurement?

As the soils from forests with well-developed organic horizons are generally acidic, we expected no presence of inorganic carbon. To confirm that, we measured the pH of soils from all sites which was in the range of 3-4.2 (Page 3, line 16). Hence, it was not necessary to remove inorganic carbon from soils.

B. What is the recovery % of short and long-chain compounds in the entire extraction process?

We employed lipid extraction procedure utilizing high temperature (100°C) and high pressure (1500 psi) accelerated solvent extraction (ASE) technique. This technique has been adopted from many previous studies, which showed > 80% recovery of leaf and soil lipid biomarkers (Ardenghi et al., 2017; Jansen et al., 2006; Magill et al., 2015). We also employed 3 extraction cycles for every sample, each cycle with static hold of pressure and temperature (5 min) and solvent flushing, which has been suggested to be sufficient for > 80% recovery of lipids. In our study, we did not calculate the absolute recovery of lipids from samples, however approx. 70% of internal standard ($C_{19:0}$ FA) was recovered from the original amount added to the test samples before lipid extraction and compound separation.

C. How the methylation correction was applied for carbon isotope ratio measurement? Which formula was used and how the alcohol isotope ratio was measured?

Correction of measured FAMEs was done by mass balance for contribution of carbon added during methyl esterification to obtain the $\delta^{13}$C value of fatty acid using following formula;

$$\delta^{13}C_{\text{FA}} = \frac{\delta^{13}C_{FAME} - (1-X)\delta^{13}C_{Methanol}}{X}$$

Where X is the ratio of carbon atoms in FA to the carbon atoms in corresponding FAME. $\delta^{13}C_{\text{Methanol}}$ was measured with an elemental analyser coupled to an isotope ratio mass spectrometer, using similar instrumental parameters as bulk C isotope analysis in our study. We added this information in the methods section to the final version of the manuscript.

**3. Result:**

A. The best way to show the result is against the respective soil profiles. The different zone identified in the different soil profiles can also be marked for clarity.

We do not clearly understand to which specific results the first point made by reviewer. However, if the reviewer suggest that the figure 5 (compound-specific isotope values) to be oriented vertically against soil horizons, we agree to the suggestion and modify it accordingly such that the trend of each compounds isotope values in the horizons placed next to each other from left to right. We will also define the organic horizons (Oi-Oe-Oa) in the method section for better clarity of the nomenclature of the horizons. Regarding the second comment, assuming it is also for figure 5, we will mark the horizon after which the isotopic signal remained stable.

B. What is LB pruce*?

$LB_{spruce*}$ is site no. 3 (Lake Baldegg catchment, Table 1) with spruce and moss as primary aboveground biomass sources and with organic horizon categorized as a raw humus. We will explain the site abbreviations in the caption of every relevant figure for better understanding.

C. At many places, you are writing d13C enrichment or depletion!! Please write an 'increase in $\delta$13C or 'enrichment of 13C'.

Thank you for the correction, we will change the wording.

D. Why moss values are neglected? Why the significantly low $\delta$13C values?

Mosses generally have lower $\delta^{13}C$ values than higher plants as they grow in wet and humid conditions and experience less water loss due to evapotranspiration. We did not neglect mosses as we did analyze their isotopic signatures. However, regarding the overall contribution to the humus layer material, we considered spruce to be a dominant source of biomass/lipids to the soils compared to moss at site $LB_{spruce*}$. Thus, for the calculation of magnitude of change in $\delta^{13}C$ value from fresh plant biomass to the mineral soil ($\Delta_1$), only needle isotope value was considered to obtain the $\delta^{13}C$ (fresh plant biomass) at site $LB_{spruce*}$. This was done to avoid overestimation of change/increase in $\delta^{13}C$, as moss isotope value is considerably depleted than that of spruce needle. Also, isotopic composition of soils is controlled by the amount of biomass from each source on a larger timescale. However, we have to admit, that we have no data to prove the dominance of needle biomass over moss biomass. We will thus formulate our interpretation a bit more cautiously.

**4. Discussion:**

A. Why the bulk $\delta$13C and is showing 13C enrichment? Compared to the nC28 and nC29 B. Is the true for other long-chain fatty acids and alkanes as well?

A+B. The majority of the bulk organic matter is considered to be less recalcitrant and more readily available to soil organisms than long-chain alkanes and fatty acids. During degradation/mineralisation, bacteria (or rather the respective enzymes) have generally a higher affinity towards the lighter isotope, which will be preferentially fed on. Hence, we observed generally a greater enrichment in $^{13}C$ of the remaining bulk organic matter across different degradation stages. A second aspect is, that previous studies have shown that lipid biomolecules are depleted in $^{13}C$ compared to bulk organic matter due to the kinetic isotope effects during various biochemical reactions in lipid biosynthesis (Chikaraishi et al., 2004; Collister et al., 1994). Hence, we already observe lower $\delta^{13}C$ values in long-chain fatty acids and *n*-alkanes compared to the bulk $\delta^{13}C$ values (Page 6, line 7-8) before degradation even starts with fractionation processes.

References

Ardenghi, N., Mulch, A., Pross, J. and Maria Niedermeyer, E.: Leaf wax n-alkane extraction: An optimised procedure, Org. Geochem., 113, 283–292, doi:10.1016/j.orggeochem.2017.08.012, 2017.

Chikaraishi, Y., Naraoka, H. and Poulson, S. R.: Carbon and hydrogen isotopic fractionation during lipid biosynthesis in a higher plant (Cryptomeria japonica), Phytochemistry, 65(3), 323–330, doi:10.1016/j.phytochem.2003.12.003, 2004.

Collister, J. W., Rieley, G., Stern, B., Eglinton, G. and Fry, B.: Compound-specific δ 13C analyses of leaf lipids from plants with differing carbon dioxide metabolisms, Org. Geochem., 21(6), 619–627, doi:10.1016/0146-6380(94)90008-6, 1994.

Jansen, B., Nierop, K. G. J., Kotte, M. C., de Voogt, P. and Verstraten, J. M.: The applicability of accelerated solvent extraction (ASE) to extract lipid biomarkers from soils, Appl. Geochem., 21(6), 1006–1015, doi:10.1016/j.apgeochem.2006.02.021, 2006.

Magill, C. R., Denis, E. H. and Freeman, K. H.: Rapid sequential separation of sedimentary lipid biomarkers via selective accelerated solvent extraction, Org. Geochem., 88, 29–34, doi:10.1016/j.orggeochem.2015.07.009, 2015.

---

## Author Response (AR1)

Reply to referee 1 comments:

Referee comments are in blue.

Abstract:

Page 1 lines 8-10: The application of CSIA is not restricted only to freshwater systems, as many studies from marine environment (For eg., Canuel et al., 1997; Limnol Oceano, 42, 1570) also show its importance

Added the relevant information in abstract and introduction, page 5 line 10 and 35.

Introduction:

Page 2 line 16: Huang et al. (1997) Punctuation- please remove comma here and many other places Page 2 line 28: I feel it should be 'fresh plant material with organic matter to the organic horizons and upper mineral soil horizon'

Removed comma page 6 line 14, page 8 line 23 and page 11 line 18,19. Restructured the sentence, page 6 line 26.

Carbon preference Index of alkanes can also be given in addition to average chain length.

CPI values were in support of our finding of general odd over even predominance in long-chain *n*-alkanes through all the degradation stages, which we have mentioned in results and discussion on Page 9 line 5. A sentence on CPI values result added on page 9 line 12.

2. Materials and methods

Page 3, line 13: It is better to define Oi-Oe-Oa horizons to the readers at the first time

Line 15: 'moder' type spelling correction

2.4 Page 4 line 7: Please define OM at the first usage Line 5: please remove the bracket for reference Schmidt et al., 1999

Oi, Oe and Oa horizons defined in study area section page 7 line 9-10. Spelling corrected. OM is defined in the introduction section. Reference corrected page 8 line 5.

Section 3.2 summarizes that bulk d13C values should not be relied for sediment source attribution compared to compound specific isotope values. However the is- sues/reasons (other than enrichment) while considering bulk isotope values are not discussed in detail.

Our aim of this study is to test if isotopic signatures of tracers used for sediment source attribution are stable which requires non-fractionation during detachment and transport. Our assumption is that if they are stable during degradation, they will also be stable during detachment and transport. Thus, the continuous enrichment

in bulk $\delta^{13}C$ values through the degradation stages is *the* reason of concluding that they are not suitable as sediment tracers. We do not know of any other major reason to reject bulk isotope analysis for sediment source attribution. Sorry, if this was not clear enough, we will have a critical look at the formulation of our aims and assumptions. We would like to add, that source apportionment with differences in $C_3$- and $C_4$- vegetation sources has been done successfully, because differences in bulk isotopic signals of sources are bigger, thus small deviations by fractionation might not substantially alter the results.

A relevant sentence is added to the results section page 10 line 10.

The manuscript is written well, but the novelty and importance of present study need to be highlighted in the manuscript.

Thank you and also for pointing out, that we should highlight our results more.

The introduction part is re-structured. In the conclusion part we added more sentences, page 12 line 8, 9, 13, and 14.

Citation of references in the text is given alphabetical order. Normally it is cited in chronological order (ascending). Anyway please follow the journal format and modify accordingly.

Citations changed to chronological order.

Inserted Upadhayay et al., (2017), see page 6 line 11 as per the suggestion.

Reply to referee 2 comments:

Referee comments are in blue.

1. Introduction:

Authors have started this section by introducing the causes of fresh ecosystem degradation as result of nutrient input along with detrital riverine/ aeolian input. However, they ended up with the determination of organic matter sources and extent of degradation in a soil horizon. The connectivity between organic matter source and concerned with freshwater ecosystems has not been addressed. This aspect needs to be elaborated in the discussion or modify the introductory part for better coherence.

We modified the introduction part, please see page 5 and 6 for marked-up changes.

2. Methodology:

A. Why inorganic carbon was not removed from the soil prior to bulk carbon isotope ratio measurement?

As the soils from forests with well-developed organic horizons are generally acidic, we expected no presence of inorganic carbon. To confirm that, we measured the pH of soils from all sites which was in the range of 3-4.2 (Page 3, line 16). Hence, it was not necessary to remove inorganic carbon from soils.

B. What is the recovery % of short and long-chain compounds in the entire extraction process?

We employed lipid extraction procedure utilizing high temperature (100°C) and high pressure (1500 psi) accelerated solvent extraction (ASE) technique. This technique has been adopted from many previous studies, which showed > 80% recovery of leaf and soil lipid biomarkers (Jansen et al., 2006; Magill et al., 2015; Ardenghi et al., 2017). We also employed 3 extraction cycles for every sample, each cycle with static hold of pressure and temperature (5 min) and solvent flushing, which has been suggested to be sufficient for > 80% recovery of lipids. In our study, we did not calculate the absolute recovery of lipids from samples, however approx. 70% of an internal standard ($C_{19:0}$ FA) was recovered from the original amount added to test samples before lipid extraction and compound separation.

C. How the methylation correction was applied for carbon isotope ratio measurement? Which formula was used and how the alcohol isotope ratio was measured?

Correction of measured FAMEs was done by mass balance for contribution of carbon added during methyl esterification to obtain the $\delta^{13}$C value of fatty acid using following formula;

$$\delta^{13}C_{FA} = \frac{\delta^{13}C_{FAME} - (1-X)\delta^{13}C_{Methanol}}{X}$$

Where X is the ratio of carbon atoms in FA to the carbon atoms in corresponding FAME. $\delta^{13}C_{Methanol}$ was measured with an elemental analyser coupled to an isotope ratio mass spectrometer, using similar instrumental parameters as bulk C isotope analysis in our study. We added this information in the methods section page 9 lines 1-5.

3. Result:

A. The best way to show the result is against the respective soil profiles. The different zone identified in the different soil profiles can also be marked for clarity.

We modified figure 4 and 5, so that the trend of each compounds' isotope values in the horizons placed next to each other from left to right. We also defined the organic horizons (Oi-Oe-Oa) in the method section. We have also marked the horizon after which the isotopic signal remained stable with a horizontal dotted line.

B. What is LB pruce*?

LB$_{spruce*}$ is site no. 3 (Lake Baldegg catchment, Table 1) with spruce and moss as primary aboveground biomass sources and with organic horizon categorized as a raw humus. We added site abbreviations in the caption of every relevant figure for better understanding.

C. At many places, you are writing d13C enrichment or depletion!! Please write an 'increase in δ13C or 'enrichment of 13C'.

Changed the wording in several places.

D. Why moss values are neglected? Why the significantly low δ13C values?

Mosses generally have lower $\delta^{13}C$ values than higher plants as they grow in wet and humid conditions and experience less water loss due to evapotranspiration. We did not neglect mosses as we did analyze their isotopic signatures. However, regarding the overall contribution to the humus layer material, we considered spruce to be a dominant source of biomass/lipids to the soils compared to moss at site LB$_{spruce*}$. Thus, for the calculation of magnitude of change in $\delta^{13}C$ value from fresh plant biomass to the mineral soil ($\Delta_1$), only needle isotope values were considered to obtain the $\delta^{13}C$ (fresh plant biomass) at site LB$_{spruce*}$. This was done to avoid overestimation of change/increase in $\delta^{13}C$, as moss isotope values are considerably depleted compared to spruce needle values. Also, isotopic composition of soils is controlled by the amount of biomass from each source on a larger timescale.

4. Discussion:

A. Why the bulk δ13C and is showing 13C enrichment? Compared to the nC28 and nC29 B. Is the true for other long-chain fatty acids and alkanes as well?

A+B. The majority of the bulk organic matter is considered to be less recalcitrant and more readily available to soil organisms than long-chain alkanes and fatty acids. During degradation/mineralisation, bacteria (or rather the respective enzymes) have generally a higher affinity towards the lighter isotope, which will be preferentially fed on. Hence, we observed generally a greater enrichment in $^{13}C$ of the remaining bulk organic matter across different degradation stages.

A second aspect is, that previous studies have shown that lipid biomolecules are depleted in $^{13}C$ compared to bulk organic matter due to the kinetic isotope effects during various biochemical reactions in lipid biosynthesis (Collister et al., 1994; Chikaraishi et al., 2004). Hence, we already observe lower $\delta^{13}C$ values in 
[revised manuscript text omitted]
\text{-}C_{24:0}$ | $n\text{-}C_{26:0}$ | $n\text{-}C_{28:0}$ | $n\text{-}C_{30:0}$ | $n\text{-}C_{25}$ | $n\text{-}C_{27}$ | $n\text{-}C_{29}$ | $n\text{-}C_{31}$ |
| BF$_{beech}$ | $\Delta_1$ | 4.0 (***) | 3.6 (**) | 3.0 (**) | 1.7 (n.s.) | 3.1 (**) | 3.2 (**) | 2.9 (**) | - |
| | $\Delta_2$ | 0.9 (n.s.) | 0.4 (n.s.) | 0.8 (n.s.) | -1.9 (n.s.) | 0.2 (n.s.) | -0.1 (n.s.) | -0.9 (n.s.) | - |
| BF$_{spruce}$ | $\Delta_1$ | 5.0 (*) | 1.8 (n.s.) | 0.2 (n.s.) | 0.2 (n.s.) | 0.9 (n.s.) | 0.6 (n.s.) | 1.1 (n.s.) | - |
| | $\Delta_2$ | 0.0 (n.s.) | -0.7 (n.s.) | -0.3 (n.s.) | -0.2 (n.s.) | -0.3 (n.s.) | 0.8 (n.s.) | 0.5 (n.s.) | - |
| BF$_{moss}$ | $\Delta_1$ | 5.2 (*) | 4.6 (n.s.) | 1.3 (n.s.) | -0.3 (n.s.) | 7.8 (**) | 5.3 (*) | 1.7 (n.s.) | 1.7 (n.s.) |
| | $\Delta_2$ | 0.3 (n.s.) | 1.1 (n.s.) | 0.7 (n.s.) | -0.3 (n.s.) | 3.9 (n.s.) | 2.4 (n.s.) | 1.4 (n.s.) | 0.6 (n.s.) |
| LB$_{spruce*}$ | $\Delta_1$ | 3.1 (n.s.) | 0.6 (n.s.) | -0.5 (n.s.) | -1.5 (n.s.) | 2.3 (n.s.) | 1.6 (n.s.) | 0.5 (n.s.) | - |
| | $\Delta_2$ | 0.8 (n.s.) | 1.0 (n.s.) | 0.3 (n.s.) | 0.2 (n.s.) | 0.5 (n.s.) | -0.3 (n.s.) | -1.3 (n.s.) | - |
| LB$_{spruce}$ | $\Delta_1$ | 3.8 (n.s.) | 1.6 (n.s.) | 0.0 (n.s.) | -0.4 (n.s.) | - | - | - | - |
| | $\Delta_2$ | 2.0 (n.s.) | 1.9 (n.s.) | -0.7 (n.s.) | -1.1 (n.s.) | 1.3 (n.s.) | 1.0 (n.s.) | 0.4 (n.s.) | -1.0 (n.s.) |
| LX$_{mixed}$ | $\Delta_1$ | -0.5 (n.s.) | -0.7 (n.s.) | -0.8 (n.s.) | -0.7 (n.s.) | 0.2 (n.s.) | 1.0 (n.s.) | 1.1 (n.s.) | -0.9 (n.s.) |
| | $\Delta_2$ | 1.5 (n.s.) | 0.8 (n.s.) | 0.9 (n.s.) | 0.6 (n.s.) | 0.8 (n.s.) | 2.2 (*) | 2.5 (**) | 1.4 (n.s.) |

Significance codes: non-significant (p > 0.05) 'n.s.'; p ≤ 0.05 '*'; p < 0.01 '**'; p < 0.001 '***'.

Abbreviations: Black Forest (BF), Lake Baldegg (LB), Upper Sûre Lake catchment-Luxembourg (LX).

[Figure]

**Figure 1: Map of all sampling sites.**

[Figure]

**Figure 2: Concentration of long-chain fatty acids and *n*-alkanes in fresh plant biomass and underlying horizons from different sites (note the different scales of fatty acid and *n*-alkane concentration). Abbreviations: Black Forest (BF), Lake Baldegg (LB), Upper Sûre Lake catchment-Luxembourg (LX).**

[Figure]

**Figure 3: Average chain length (ACL) of fatty acids (ACL$_{22-32}$) and *n*-alkanes (ACL$_{21-33}$). Abbreviations: Black Forest (BF), Lake Baldegg (LB), Upper Sûre Lake catchment-Luxembourg (LX).**

[Figure]

**Figure 4: Bulk δ$^{13}$C values (‰) of fresh plant biomass (leaves, needles & mosses) and underlying horizons. Abbreviations: Black Forest (BF), Lake Baldegg (LB), Upper Sûre Lake catchment-Luxembourg (LX).**

[Figure]

**Figure 5: Compound-specific δ¹³C values of long-chain fatty acids ($n$-C$_{24:0}$ to $n$-C$_{30:0}$) and long-chain $n$-alkanes ($n$-C$_{25}$ to $n$-C$_{31}$) in fresh plant biomass (leaves, needles & mosses) and across the underlying horizons (mean ± measurement error of 3 replicates).** Black Forest (beech (1), spruce (2a), moss (2b)), Lake Baldegg (spruce* (3), spruce (4)), Upper Sûre Lake catchment-Luxembourg (mixed forest (5)). Abbreviations: Fatty acids (FA), $n$-Alkanes (Alk). δ¹³C values generally remained stable after the horizon indicated by a horizontal dotted line.

[Figure]

**Figure 6: Comparison of the δ13C values from two particle-size fractions in soil from four sites (mean ± measurement error of 3 replicates). Abbreviations: Black Forest (BF), Lake Baldegg (LB), Upper Sûre Lake catchment-Luxembourg (LX).**